# Wavelet-Based Entropy Measures to Characterize Two-Dimensional Fractional Brownian Fields

**DOI:** 10.3390/e22020196

**Published:** 2020-02-07

**Authors:** Orietta Nicolis, Jorge Mateu, Javier E. Contreras-Reyes

**Affiliations:** 1Facultad de Ingenieria, Universidad Andres Bello, Viña del Mar 2520000, Chile; 2Department of Mathematics, Universitat Jaume I, E-12071 Castellon, Spain; 3Departamento de Estadistica, Universidad del Bio-Bio, Concepcion 4081112, Chile

**Keywords:** fractional Brownian motion, Rényi entropy, Shannon entropy, Tsallis entropy, wavelets

## Abstract

The aim of this work was to extend the results of Perez et al. (Physica A (2006), 365 (2), 282–288) to the two-dimensional (2D) fractional Brownian field. In particular, we defined Shannon entropy using the wavelet spectrum from which the Hurst exponent is estimated by the regression of the logarithm of the square coefficients over the levels of resolutions. Using the same methodology. we also defined two other entropies in 2D: Tsallis and the Rényi entropies. A simulation study was performed for showing the ability of the method to characterize 2D (in this case, α=2) self-similar processes.

## 1. Introduction

The concept of entropy was first introduced by [1] in thermodynamics as a measure of the amount of energy in a system. Posteriorly, Boltzmann [2] was the first who gave a probabilistic interpretation, setting the foundations of statistical physics. Shannon [3] proposed the entropy concept on the subject of information theory as the average rate at which information is produced by a stochastic data source. According to information theory, entropy is a measure of uncertainty and unpredictability associated with a random variable (discrete or continuous), and Shannon entropy quantifies the expected value of information generated from a random variable. The definition of entropy has been widely used in many applications, such as neural systems [4], image segmentation through thresholding [5,6], climatology, and hydrology [7,8,9,10]. Nicholson et al. [11] introduced spatial entropy to study earthquake distributions. A temporal definition of entropy was used by [12,13,14,15] to study seismicity in different parts of the world.

For a process characterized by a certain number *N* of states or classes of events, Shannon entropy [3] is defined as
(1)$$S=-\sum_{i=1}^{N}p_{i}\log\left(p_{i}\right),$$
where $p_{i}$ is the probability of event occurrence in each *i*-th class. The choice of the base of the logarithm is arbitrary: for practical convenience, we used base two throughout this paper ($\log\equiv \log_{2}$). For $p_{i}=0$, $p_{i}\log_{2}p_{i}=0$. Shannon entropy is maximal when all outcomes are equally likely, that is, $S=\log_{2}\left(N\right)$.

A generalization of Shannon entropy is Rényi entropy proposed by [16] as
(2)$$S_{\alpha }=\frac{1}{1-\alpha }\log_{2}\left(\sum_{i=1}^{N}p_{i}^{\alpha }\right),$$
where α, α>1, is a parameter. Shannon entropy is obtained from $S_{\alpha }$ as α→1 (see, e.g., [17]), and Rényi entropy is non-negative in discrete Case (Equation 2). Finally, for any $\alpha_{1}<\alpha_{2}$, we have $S_{\alpha_{1}}\geq S_{\alpha_{2}}$ (and $S_{\alpha_{1}}=S_{\alpha_{2}}$ if and only if the system is uniformly distributed).

Another generalization of Shannon entropy is Tsallis entropy, proposed by [18] as
(3)$$T_{\alpha }=\frac{1}{\alpha -1}\left(1-\sum_{i=1}^{N}p_{i}^{\alpha }\right).$$
Tsallis entropy satisfies the following properties: (i) $T_{\alpha }\geq 0$ with $\left(\alpha -1\right)T_{\alpha }\leq 1$; (ii) $T_{\alpha }\to S$ as α→1; (iii) pseudoadditivity between two independent systems *A* and *B*: $T_{\alpha }\left(A,B\right)=T_{\alpha }\left(A\right)+T_{\alpha }\left(B\right)+\left(1-\alpha \right)T_{\alpha }\left(A\right)T_{\alpha }\left(B\right)$ (additivity is accomplished if α=1); and (iv) it is a non-decreasing function of the Rényi entropy because [19]
$$S_{\alpha }=\frac{1}{1-\alpha }\log_{2}\left[1+\left(1-\alpha \right)T_{\alpha }\right].$$

However, probability density is not the only type of distribution that can give information, and the definition of entropy can be extended to other types of distributions, such as energy distribution based on wavelet coefficients [20]. A definition of Shannon wavelet entropy based on the energy distribution of wavelet coefficients was proposed by [21,22,23,24,25,26,27]. In particular, Sello [21] defined temporal wavelet entropy using continuous wavelets, and [20] introduced multiresolution wavelet entropy by summing the energy for all discrete times, and discretizing scale *j*. Nicolis and Mateu [28] used the anisotropic Morlet wavelet to define Shannon entropy in two-dimensional point processes. A discrete version of Shannon wavelet entropy was proposed by [25,26,27,29,30,31,32] to characterize self-similar processes with Gaussian and stationary increments. Recently, a similar approach based on wavelet probability densities was proposed by [33] using the Fisher–Shannon method [15].

However, the latter authors used a definition of wavelet entropy for characterizing self-similar processes in the time domain, but an extension to the two-dimensional case has not been proposed so far. In this work, we extend the definition of wavelet Shannon entropy proposed by [25,26,27] to provide a characterization of an isotropic *n*-dimensional fractional Brownian field. The same methodology is also used for defining wavelet-based Rényi and Tsallis entropies. A simulation study is provided to prove and check the results.

The article is organized as follows: Section 2 provides some basic concepts of fractional Brownian motion and its extensions. Section 3 gives some definitions of 2D wavelet transforms. Directional wavelets and anisotropic wavelet entropy are introduced in Section 4. A simulation study is reported in Section 5. The paper ends with some conclusions in Section 6.

## 2. Fractional Brownian Motion and Extensions

Fractional Brownian motion (fBm) denoted by $\left\{B_{H}\left(t\right),t\in R\right\}$, is a Gaussian, zero-mean, nonstationary stochastic process originally proposed by [34]. This process is called self-similar since, for all a>0, it satisfies
$$B_{H}\left(at\right)\overset{d}{=}a^{H}B_{H}\left(t\right),$$
where *H* is the self-similarity or Hurst exponent parameter, and “$\overset{d}{=}$” denotes equality in distribution. The fBm process is characterized by the following covariance function:(4)$$R_{B_{H}}\left(t,s\right)=E\left\{B_{H}\left(t\right)B_{H}\left(s\right)\right\}=\frac{\sigma_{H}^{2}}{2}\left[{\left|t\right|}^{2H}+{\left|s\right|}^{2H}-{\left|t-s\right|}^{2H}\right],$$
where
$$\sigma_{H}^{2}=\Gamma \left(1-2H\right)\frac{\cos(\pi H)}{\pi H},\;0<H<1.$$

As can be seen from Function (Equation 4), the fBm is a nonstationary process, but with stationary increments. These definitions can be extended to any dimension. The case of fBm generalization from one to higher dimensions is not unique. A simple generalization to a 2D surface is the fractional Brownian field (fBf). The fBf is a Gaussian, zero-mean, random field $B_{H}\left(u\right)$, where u denotes the position in a selected domain, usually [0,1]×[0,1]. Its covariance function is given by
(5)$$R_{B_{H}}\left(u,v\right)=E\left[B_{H}\left(u\right)B_{H}\left(v\right)\right]=\frac{\sigma_{H}^{2}}{2}\left({∥u∥}^{2H}+{∥v∥}^{2H}+{∥u-v∥}^{2H}\right),$$
where 0<H<1, the variance $\sigma_{H}^{2}$ is
(6)$$\sigma_{H}^{2}=\frac{2^{-(1+2H)}\Gamma \left(1-H\right)}{\pi H\Gamma (1+H)},$$
and $∥\cdot ∥$ is the usual Euclidean norm in $R^{2}$ (see [35,36,37]). The increments of an fBf represent stationary, zero-mean, Gaussian random fields because the variance of such increments only depends on distance $∥h∥$ so that
$$E{\left[B_{H}\left(u+h\right)-B_{H}\left(u\right)\right]}^{2}=\sigma_{H}^{2}{∥h∥}^{2H},$$
where $\sigma_{H}^{2}$ is given in Equation (Equation 6).

The extension to the *d*-dimensional case is straightforward [38]. For a *d*-dimensional fractional Brownian motion, the covariance function is given by Equation (Equation 5) with u,v in $R^{d}$, and
(7)$$\sigma_{H}^{2}=\frac{2^{-1-d-2H}\Gamma \left(1-H\right)}{\pi^{\frac{d}{2}}H\Gamma \left(\frac{d}{2}+H\right)}.$$

Although many generalizations have been proposed to include anisotropy in Gaussian random fields [37,39], in the following section, we only consider the isotropic version of the fBf.

## 3. Two-Dimensional Wavelet Transforms

In one or higher dimensions, wavelets provide an appropriate tool for analyzing self-similar signals or objects. In the two-dimensional domain, wavelet transforms can be constructed through translations and the dyadic scaling of a product of univariate wavelets and scaling functions. Using the same setting as that provided in [35], the following so-called separable 2D wavelets can be defined as
$$\begin{array}{cc} \phi (u_{x},u_{y}) & =\phi \left(u_{x}\right)\cdot \phi \left(u_{y}\right),\\ \psi^{h}\left(u_{x},u_{y}\right) & =\phi \left(u_{x}\right)\cdot \psi \left(u_{y}\right),\\ \psi^{v}\left(u_{x},u_{y}\right) & =\psi \left(u_{x}\right)\cdot \phi \left(u_{y}\right),\\ \psi^{d}\left(u_{x},u_{y}\right) & =\psi \left(u_{x}\right)\cdot \psi \left(u_{y}\right), \end{array}$$
where ϕ and ψ are scaling and wavelet functions, and symbols h,v,d stand for the horizontal (*h*), vertical (*v*), and diagonal (*d*) directions, respectively. So, any function $g\in L^{2}\left(R^{2}\right)$ has the following representation:(8)$$g\left(u\right)=\sum_{k=(k_{1},k_{2})}c_{j_{0},k}\phi_{j_{0},k}\left(u\right)+\sum_{j\geq j_{0}}\sum_{k=(k_{1},k_{2})}\sum_{i=h,v,d}d_{j,k}^{i}\psi_{j,k}^{i}\left(u\right),$$
with $u=\left(u_{x},u_{y}\right)\in R^{2}$, $\left(k_{1},k_{2}\right)\in Z^{2}$, $\phi_{j,k}\left(u\right)$ and $\psi_{j,k}\left(u\right)$ being the translations and dilations of the scaling function and of the mother wavelet, defined by
$$\begin{array}{ccc} \phi_{j,k}\left(u\right) & = & 2^{2j}\phi \left(2^{j}u_{x}-k_{1},2^{j}u_{y}-k_{2}\right),\\ \psi_{j,k}^{i}\left(u\right) & = & 2^{2j}\psi^{i}\left(2^{j}u_{x}-k_{1},2^{j}u_{y}-k_{2}\right), \end{array}$$
for i=h,v,d [40]. The scaling and detail coefficients in Equation (Equation 8), respectively, are given by
$$\begin{array}{cc} c_{j,k} & =\langle g,\phi_{j,k}\rangle =2^{2j}\int_{{I\;R}^{2}}g\left(t\right)\phi \left(2^{j}t-k\right)dt,\\ d_{j,k}^{i} & =\langle g,\psi_{j,k}\rangle =2^{2j}\int_{{I\;R}^{2}}g\left(t\right)\psi^{i}\left(2^{j}t-k\right)dt, \end{array}$$
for i=h,v,d. With this notation, $j_{0}$ denotes the coarsest scale and therefore the lowest resolution in the representation. A larger *j* corresponds to a finer scale, and therefore corresponds to a higher resolution. If M×M is the size of the matrix representing a 2D object (for example, an image), the number of coefficients for each level of resolution and direction *i* is $2^{j}M\times 2^{j}M$ with j=−N,…,−1 and $N=\log_{2}\left(M\right)$, where *M* must be taken a priori to be an integer power of 2, i.e., $M=2^{N}$. For further details on wavelet theory, see [40,41]. Wavelet transform can be also extended to the n−dimensional case (see, for example, [42] for a 3D case).

## 4. Shannon Wavelet Entropy for 2D FBF

In this section, in order to address Shannon wavelet entropy for a 2D fBf, we deal with a random signal (with some second-order properties) for wavelet energy to be written in terms of expectations. Following the notation of [26] for the one-dimensional case, 2D wavelet energy at resolution *j* can be written as
$$E^{i}\left(j\right)=\sum_{k=(k_{1},k_{2})}E[|d_{j,k}^{i}{|}^{2}],$$
and 2D relative wavelet energy (RWE) is given by
(9)$$p^{i}\left(l\right)=\frac{E^{i}\left(l\right)}{\sum_{l=-N}^{-1}E^{i}\left(l\right)}=\frac{\sum_{k=(k_{1},k_{2})}E[|d_{l,k}^{i}{|}^{2}]}{\sum_{l=-N}^{-1}\sum_{k=(k_{1},k_{2})}E[|d_{l,k}^{i}{|}^{2}]},$$
for varying index *l*, l=−N…−1, being *N* is the maximal resolution level.

Consequently, 2D normalized Shannon, Rényi, and Tsallis wavelet entropies (NSWE, NRWE, and NTWE, respectively) can be defined as
(10)$$\begin{array}{ccc} S^{i} & = & -\frac{1}{\log_{2}N}\sum_{l=-N}^{-1}p^{i}\left(l\right)\log_{2}\left(p^{i}\left(l\right)\right), \end{array}$$
(11)$$\begin{array}{ccc} S_{\alpha }^{i} & = & \frac{\log_{2}\left(\sum_{l=-N}^{-1}{\left[p^{i}\left(l\right)\right]}^{\alpha }\right)}{\left(1-\alpha \right)\log_{2}N}, \end{array}$$
(12)$$\begin{array}{ccc} T_{\alpha }^{i} & = & \frac{1-\sum_{l=-N}^{-1}{\left[p^{i}\left(l\right)\right]}^{\alpha }}{\left(\alpha -1\right)\log_{2}N}, \end{array}$$
respectively.

For an fBf process $B_{H}\left(x\right)$, the detail coefficients are random variables given by
(13)$$d_{j,k}^{i}=2^{j}\int_{{I\;R}^{2}}B_{H}\left(x\right)\psi^{i}\left(2^{j}x-k\right)dx,$$
where i=h,v or *d*. The detail coefficients have zero mean and variance (see [35,43]) given by
(14)$$E[|d_{j,k}^{i}{|}^{2}]=2^{2j}\int_{{I\;R}^{2}}\;\;\;\int_{{I\;R}^{2}}\psi^{i}\left(2^{j}x-k\right)\psi^{i}\left(2^{j}v-k\right)E\left[B_{H}\left(x\right)B_{H}\left(v\right)\right]dxdv.$$

From Equation (Equation 14), we can derive
(15)$$E[|d_{j,k}^{i}{|}^{2}]=\frac{\sigma_{H}^{2}}{2}V_{\psi^{i}}2^{-(2H+2)j},$$
where
(16)$$V_{\psi^{i}}=-\int_{{I\;R}^{2}}\;\;\;\int_{{I\;R}^{2}}\psi^{i}\left(p+q\right)\cdot \psi^{i}\left(q\right){\left|p\right|}^{2H}dpdq$$
with p=2j(x−v) and $q=2^{j}v-k$ (see [35], for the derivation of this result). (Equation 16) only depends on wavelets $\psi^{i}$ and exponent *H*, but not on scale *j*.

An application of the logarithm with base two to both sides of Expression (Equation 15) leads to the following equation:(17)$$\log_{2}E[|d_{j,k}^{i}{|}^{2}]=-\left(2H+2\right)j+C_{i},$$
where
$$C_{i}=\log_{2}\left\{\frac{\sigma_{H}^{2}}{2}V_{\psi^{i}}\left(H\right)\right\}.$$

The Hurst coefficient of an fBf is estimated from the slope of the linear equation given in Equation (Equation 17). The empirical counterpart of Equation (Equation 17) is regression defined in pairs,
(18)$$\left(j,\log_{2}\bar{{\left|d_{j,k}^{i}\right|}^{2}}\right),\;i=h,v,d,$$
where $\bar{{\left|d_{j,k}^{i}\right|}^{2}}$ is an empirical counterpart of $E\left[{\left|d_{j,k}^{i}\right|}^{2}\right]$ [35].

By replacing Equation (Equation 15) into 2D RWE (Equation 9), we obtain the relative wavelet energy at direction *i* for a 2D fractional Brownian field, that is,
(19)$$p^{i}\left(l\right)=\frac{2^{-(2H+2)l}}{\sum_{l=-N}^{-1}2^{-(2H+2)l}},$$
for varying index *l*.

Since for isotropic processes, relative wavelet energy is independent of wavelet basis, $p^{i}\left(j\right)$ are equal for each direction i=d,h,v. However, it could be not true for general isotropic processes without the self-similarity condition. Next, we only considered the d−direction, and we denote by p(j) its relative wavelet energy at each resolution *j*.

**Proposition** **1.**
*Let H be the Hurst exponent parameter and N the number of resolution levels, then:*
$$\sum_{j=-N}^{-1}2^{-j(2+2H)}=2^{\left(2+2H\right)}\left(\frac{1-2^{-(2+2H)N}}{1-2^{-(2+2H)}}\right).$$


For proof of Proposition 1, see the Appendix A. For a 2D object of M×M size, and with a maximal resolution level already explicitly fixed to $N=\log_{2}\left(M\right)$, Equation (Equation 19) can be written using Proposition 1 as
(20)$$p\left(j\right)=\left(\frac{1-2^{-(2+2H)}}{1-2^{-(2+2H)N}}\right)2^{-(2+2H)(j+1)}.$$

Similarly to the 1D case described in [27], by replacing Equation (Equation 19) in Equation (Equation 10), and considering γ=1 in Proposition 1, wavelet-based Shannon entropy for a grid-sampled fBf and fixing maximal resolution level *N* is given by
(21)$$S\left(N,H\right)=\frac{1}{\log_{2}\left(N\right)}\left(2+2H\right)\left[\frac{1}{2^{\left(2+2H\right)}-1}-\frac{N}{2^{(2+2H)N}-1}\right]-\frac{1}{\log_{2}\left(N\right)}\log_{2}\left[\frac{1-2^{-(2+2H)}}{1-2^{-(2+2H)N}}\right],$$
which only depends on *H* and *N*. Equation (Equation 21) can be easily generalized to *m* dimensions by using
$$E\left[{\left|d_{j,k}^{i}\right|}^{2}\right]=2^{-(2H+m)}\sigma_{H}^{2}$$
in Equation (Equation 20), with
$$\sigma_{H}^{2}=\Gamma \left(1-2H\right)\frac{\cos(\pi H)}{\pi H},\;0<H<1.$$

**Proposition** **2.**
*Let H be the Hurst exponent parameter, N the number of resolution levels, and α>1; then,*
$$\sum_{j=-N}^{-1}{\left[p^{i}\left(j\right)\right]}^{\alpha }={\left(\frac{1-2^{-(2+2H)}}{1-2^{-(2+2H)N}}\right)}^{\alpha }\left(\frac{1-2^{-\alpha (2+2H)N}}{1-2^{-\alpha (2+2H)}}\right).$$


For a proof of Proposition 2, see the Appendix A. By replacing Equation (Equation 19) in Equations (11) and (12), and considering Proposition 2, wavelet-based Rényi and Tsallis entropies for a grid-sampled fBf and fixing maximal resolution level *N* are given by
(22)$$\begin{array}{ccc} S_{\alpha }\left(N,H\right) & = & \frac{1}{\left(1-\alpha \right)\log_{2}N}\log_{2}\left\{{\left(\frac{1-2^{-(2+2H)}}{1-2^{-(2+2H)N}}\right)}^{\alpha }\left(\frac{1-2^{-\alpha (2+2H)N}}{1-2^{-\alpha (2+2H)}}\right)\right\}, \end{array}$$
(23)$$\begin{array}{ccc} T_{\alpha }\left(N,H\right) & = & \frac{1}{\left(\alpha -1\right)\log_{2}N}\left\{1-{\left(\frac{1-2^{-(2+2H)}}{1-2^{-(2+2H)N}}\right)}^{\alpha }\left(\frac{1-2^{-\alpha (2+2H)N}}{1-2^{-\alpha (2+2H)}}\right)\right\}, \end{array}$$
respectively; both only depend on α, *H*, and *N*.

## 5. Simulation Study

For illustrative purposes, we simulated N=1000 2D fractional Brownian fields with 1024×1024 size and *H* ranging from 0.1 to 0.9. Figure 1 represents two simulated 2D fractional Brownian fields with H=0.3 and H=0.8, respectively. For each simulation, we estimatd the Hurst parameter *H* and wavelet-based Shannon entropy S(N,H) using wavelet Daubechies 6. The boxplot represented in Figure 2 shows the distribution of the Hurst parameter estimation for each value of *H* (for H=0.1,…,0.9).

Results were similar to those obtained by [27] for the one-dimensional case: the wavelet-based estimator had good performance for H>0.3. In Figure 3, wavelet-based Shannon entropy measures are compared with the theoretical result of Equation (Equation 21) using H=0.1,…,0.9. In particular, in Figure 3a, we used the empirical wavelet coefficients for estimating wavelet-based entropy, such as in Equation (Equation 10). In Figure 3b, we used the estimated Hurst parameters for estimating the wavelet-based Shannon entropy of Equation (Equation 21).

Although the variability of wavelet-based entropy was higher in the empirical case, median values were very close to the theoretical results. This result can be used for characterizing a 2D fractional Brownian field and describing its entropy.

Additionally, we estimated wavelet-based Tsallis and Rényi (for α=2) entropies for the simulated 2D fractional Brownian fields (Figure 4a,b, respectively) and we compared them with their theoretical values for each *H* (with H=0.1,…,0.9). These plots show that Tsallis and Rényi entropies both decreased with higher values of *H*. However, more robust methods have to be used for estimating Hurst parameters when H is very small (for example, for H=0.1), since the wavelet method seems to underestimate real values, hence affecting the estimation of entropy measures. Since the result of this work allows to estimate entropy measures independently of the used wavelets, a different method could be used for estimating the Hurst parameter. In Figure 5a,b, we show the different behavior of Tsallis and Rényi entropies, respectively, for different values of α.

## 6. Conclusions

In this work, we derived a mathematical expression for defining 2D Shannon, Tsallis, and Rényi entropies for a 2D fractional Brownian field. Results showed that the different proposed entropies are independent from the choice of wavelet function, allowing the use of different methods for estimating the Hurst parameter. The proposed formulations could be used in many applications where the generating process is a 2D fractional Brownian motion. Furthermore, these results could easily be extended to the n− dimensional case. Some generalizations could also be considered for the study of anisotropic fractional Brownian fields by taking into account continuous wavelet transform, such as fully anisotropic wavelets introduced by [44] and successively used by [28]. Finally, our future steps are evaluating and validating these results on real datasets.

## Figures and Tables

**Figure 1 entropy-22-00196-f001:**
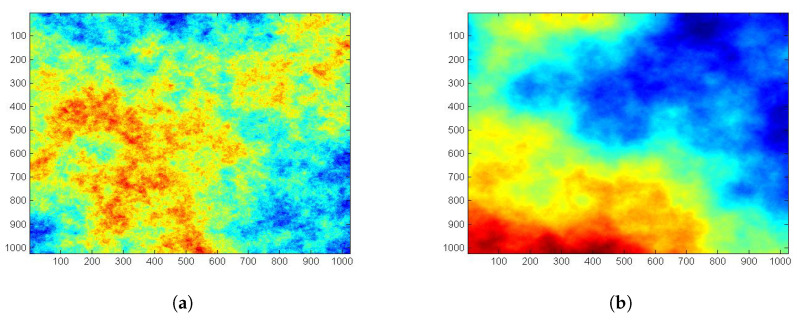
Simulated fractional Brownian fields (fBf) using (**a**) H=0.3 and (**b**) H=0.8.

**Figure 2 entropy-22-00196-f002:**
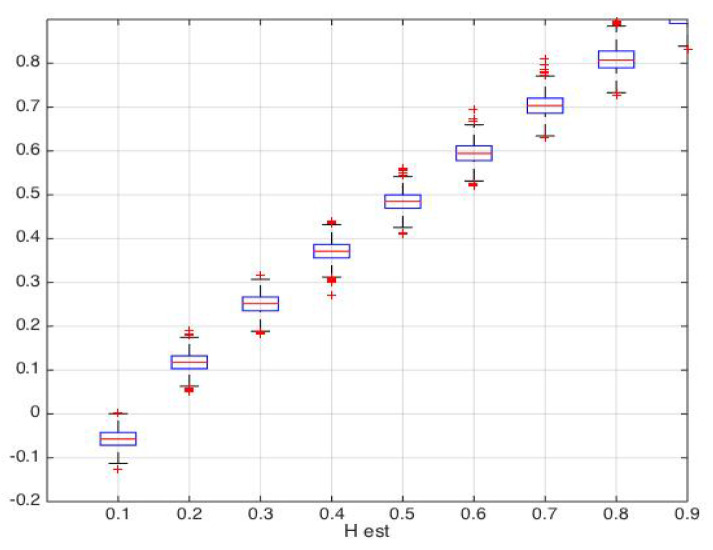
Boxplots of estimated Hurst parameter for each *H* parameter (H=0.1,…,0.9). Dashed line, identity.

**Figure 3 entropy-22-00196-f003:**
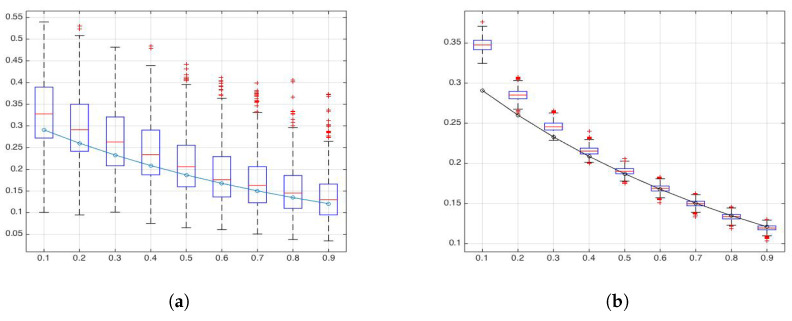
(**a**) Boxplots of empirical wavelet-based Shannon wavelet entropy and (**b**) theoretical wavelet-based Shannon entropy using estimated Hurst parameters. Dashed line, theoretical wavelet Shannon entropy S(8,H) for H=0.1,…,0.9.

**Figure 4 entropy-22-00196-f004:**
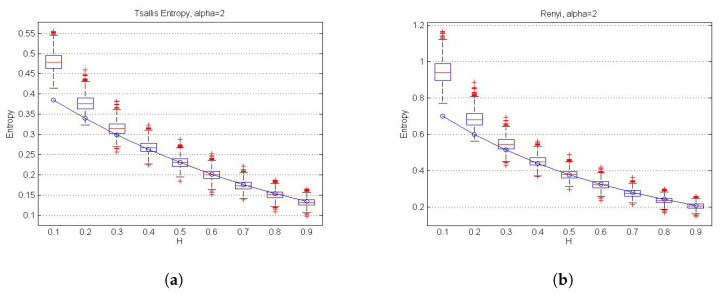
Boxplots of theoretical wavelet-based (**a**) Tsallis and (**b**) Rényi entropies using estimated Hurst parameters. Dashed line, theoretical Tsallis ($T_{\alpha }\left(N,H\right)$) and Rényi ($S_{\alpha }\left(N,H\right)$) entropies, respectively, for α=2 and H=0.1,…,0.9.

**Figure 5 entropy-22-00196-f005:**
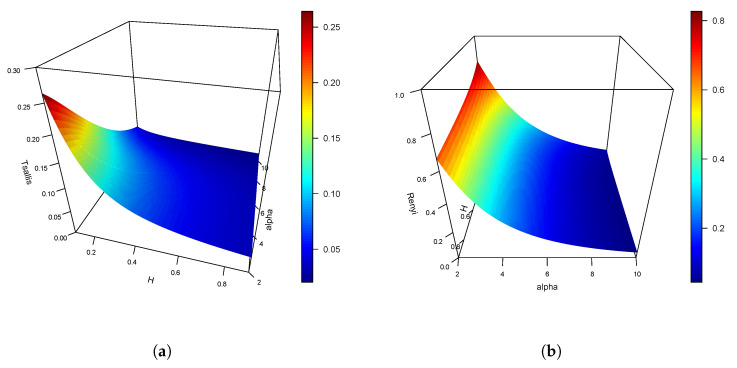
Theoretical wavelet-based (**a**) Tsallis and (**b**) Rényi entropies for N=9, H=0.1,…,0.9, and α>2.

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
