# Peer review of "Wavelet-Based Entropy Measures to Characterize Two-Dimensional Fractional Brownian Fields"

_entropy, 2020, doi:10.3390/e22020196_

Round 1

Reviewer 1 Report

Please find attached report.

Reviewer 2 Report

The paper is of some interest, but the mathematics is not clear and seems to be incorrect, and English is sometimes poor.

Reviewer 3 Report

1.- The paper presents many English typos and the authors should make a revision of their English language. In the abstract “Tsallys” should be changed to “Tsallis” since it is the correct term.

2.- In the abstract the authors should clarify whether the Shannon entropy is computed using the Hurst exponent… or by using the wavelet spectrum (from which the Hurst exponent is estimated).

3.- The references should be cited by appearance, e.g., first citation “The concept of entropy was first introduced by [3]” should be changed to “The concept of entropy was first introduced by [1]” and listed accordingly in the references section.

4.- The reference in line 21 is omitted and that’s why it appears as “?”. Please correct this typo.

5.- In line 36 the authors claim “However the authors proposed a …… in the time domain.”, however the correct claim should be based in that the above authors used a definition based on a one-dimensional metric and the proposed paper in the n-dimensional case.

6.- It would be interesting to see the behaviour of the different plots in their Tsallis or Rényi plots.

7.- In the same way references [9] and [25][26] provide illustrative information of wavelet Shannon entropy it would be interesting to cite

Pérez, D. G., Zunino, L., Martin, M. T., Garavaglia, M., Plastino, A., & Rosso, O. A. (2007). Model-free stochastic processes studied with q-wavelet-based informational tools. Physics Letters A364(3-4), 259-266.

Ramírez-Pacheco, J., Rizo-Domínguez, L., Trejo-Sánchez, J. A., & Cortez-González, J. (2016). A nonextensive wavelet (q, q´)-entropy for 1/?α signals. Revista mexicana de física62(3), 229-234.

Zunino, L., Pérez, D. G., Kowalski, A., Martín, M. T., Garavaglia, M., Plastino, A., & Rosso, O. A. (2008). Fractional Brownian motion, fractional Gaussian noise, and Tsallis permutation entropy. Physica A: Statistical Mechanics and its Applications387(24), 6057-6068.

Ramirez Pacheco, J., Torres Román, D., & Toral Cruz, H. (2012). Distinguishing Stationary/Nonstationary Scaling Processes Using Wavelet Tsallis-Entropies. Mathematical Problems in Engineering2012.

Since it includes additional information on wavelet Tsallis entropy.

Round 2

Reviewer 2 Report

Now the paper is carefully revised and current version is satisfactory. However, for the future research my advice to the authors is to study more mathematical papers devoted to multi-parameter multi-dimensional fractional fields, for example, the papers devoted to the existence of local time, in order to distinguish the difference more precisely.